# Patients with basal ganglia damage show preserved learning in an economic game

Lusha Zhu [1], Yaomin Jiang [1], Donatella Scabini[2], Robert T. Knight[2,3] & Ming Hsu [2,4]

Both basal ganglia (BG) and orbitofrontal cortex (OFC) have been widely implicated in social and non-social decision-making. However, unlike OFC damage, BG pathology is not typically associated with disturbances in social functioning. Here we studied the behavior of patients with focal lesions to either BG or OFC in a multi-strategy competitive game known to engage these regions. We find that whereas OFC patients are significantly impaired, BG patients show intact learning in the economic game. By contrast, when information about the strategic context is absent, both cohorts are significantly impaired. Computational modeling further shows a preserved ability in BG patients to learn by anticipating and responding to the behavior of others using the strategic context. These results suggest that apparently divergent findings on BG contribution to social decision-making may instead reflect a model where higher-order learning processes are dissociable from trial-and-error learning, and can be preserved despite BG damage.

[1] School of Psychological and Cognitive Sciences; Beijing Key Laboratory of Behavior and Mental Health; IDG/McGovern Institute for Brain Research; Peking-Tsinghua Center for Life Sciences, Peking University, Beijing 100871, China. [2] Helen Wills Neuroscience Program, University of California, Berkeley, CA 94720, USA. [3] Department of Psychology, University of California, Berkeley, CA 94720, USA. [4] Haas School of Business, University of California, Berkeley, CA 94720, USA. Correspondence and requests for materials should be addressed to L.Z. (email: lushazhu@pku.edu.cn) or to M.H. (email: mhsu@haas.berkeley.edu)

Guided by dopaminergic inputs from the substantia nigra, the basal ganglia (BG), along with other regions of the frontostriatal circuits including the orbitofrontal cortex (OFC), have been widely implicated in value-based decision-making involving learning and instantiation of behavioral policies[1,2]. In recent years, there is growing evidence, primarily from neuroimaging, suggesting that these regions may in addition play a crucial role in reward-guided behavior in the social domain[3–5]. In particular, studies applying formal computational models of social decision-making have begun to elucidate cognitive mechanisms underlying putative BG and OFC contributions to social valuation and learning processes, paralleling their roles in more basic decisions involving rewards and punishments[3,6–9].

Owing to the inherently correlational nature of functional neuroimaging measures, however, there is substantial uncertainty regarding the specific causal contribution of BG to social behavior. Unlike in OFC patients where social and emotional disturbances have long been a hallmark of damage[10], BG pathology is not typically associated with social dysfunction[11–13]. Clinical reports of focal BG lesion include few observations of social deficits[11]. Studies of patients where damage was acquired through neurodegenerative disorders such as Parkinson's disease have likewise yielded mixed results, where social functioning deficits manifest primarily in late stage patients, when affected regions likely include prefrontal areas[14–16]. Indeed, this is consistent with growing evidence that there exists a multiplicity of decision-making and learning mechanisms that are supported by dissociable neural systems, and whose interaction is critical for understanding their causal contribution to behavioral outputs[6,7,17]. Thus, it is possible that social functioning is spared at least to some extent in BG pathologies owing to compensatory mechanisms supported by intact regions along the frontostriatal circuits. Alternatively, it is possible that social deficits were overlooked or underreported in previous studies owing to more visible deficits associated with BG pathology, such as those involving motor functioning[11].

Here we sought to test the above hypotheses by comparing the behavior of patients with focal lesion to either the BG or OFC to that of healthy compare subjects in a multi-strategy competitive game, the so-called Patent Race, where the success of players' actions depend on those of coplayers[7,18] (Fig. 1; Methods). Unlike cortical regions, which can be reached using noninvasive brain stimulation methods such as transcranial magnetic stimulation (TMS), patient studies remain one of the few methods available to human researchers to gain causal insights into the functional role of BG to cognition and behavior[19–24]. Specifically, we employed a stylized but well-characterized setting of a population with many anonymously interacting agents and low probability of re-encounter (Methods). This setting provides a natural model for situations such as commuters in traffic or bargaining in bazaars.

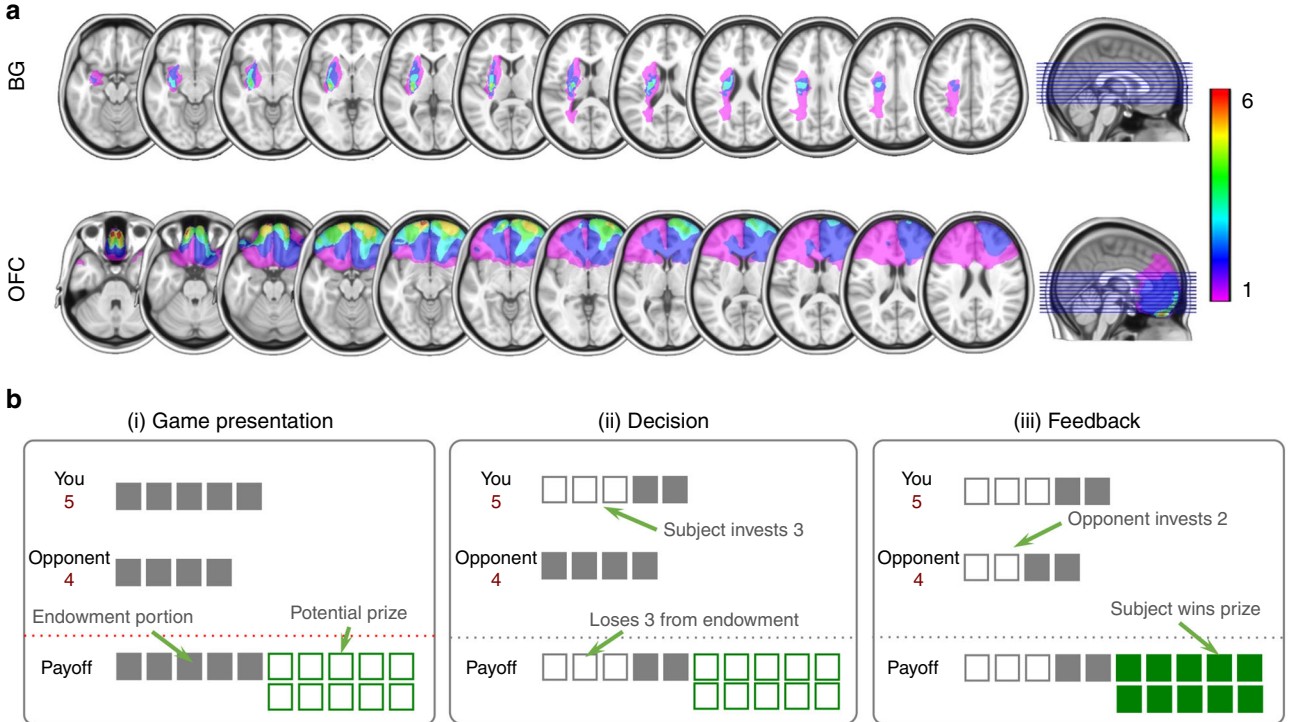

**Fig. 1** Lesion reconstruction and task schematic. **a** Structural MRI slices illustrating the lesion overlap across the two patient groups. All BG lesions were shown overlaid on the left hemisphere for comparison purposes (4L; 2R). BG group mean lesion volume was 10.6 cm³. Maximal lesion overlap was in the putamen and encompassed the head and body of the caudate as well as the globus pallidus in some patients. OFC group mean lesion volume was 113.5 cm³. Maximal lesion overlap was in Brodmann's areas 10, 11, 13, and 14, centered in the OFC and including portions of areas 12, 25, and 47 in some patients. **b** Subjects were presented with the information regarding their endowment, the endowment of the opponent, and the potential prize. In the particular payoff structure we used, the prize is worth 10 units, and the Strong (Weak) player is endowed with five (4) units at the beginning of each round. Subjects then inputted the decision (self-paced) by pressing a button mapped to the desired investment amount from the initial endowment. If the subject's investment was strictly more than that of the opponent, the subject won the prize; otherwise, the subject lost the prize. In the event of a tie, both lost the prize. In either case, the subject kept the portion of the endowment not invested. In the non-strategic condition, participants were told to choose an investment that must exceed a randomly generated hurdle to win the prize, but were not told how the computer generated the random hurdle. The hurdle followed the same empirical frequency of decisions as in the strategic condition. The experiment consisted of 160 rounds of Patent Race game, alternating between the strategic and non-strategic conditions over 80 rounds, counterbalanced

Importantly, in minimizing the role of reputation and higher-order belief considerations, the population setting using a random matching protocol is perhaps the most widely studied experimental setting and has served as a basic building block for a number of models in evolutionary biology and game theory[7,25–27].

Specifically, in the Patent Race, players of two types, Strong and Weak, are randomly matched at the beginning of each round and compete for a prize by choosing an investment (in integer amounts) from their respective endowments. The player who invests more wins the prize, and the other loses. In the event of a tie, both lose the prize. Regardless of the outcome, players lose the amount that they invested. In the particular payoff structure we use, the prize is worth 10 units, and the Strong (Weak) player is endowed with 5 (4) units (Fig. 1b).

Substantial evidence has shown that learning in economic games including the Patent Race can be parsimoniously explained using two learning rules across a wide-range of strategic contexts and experimental conditions: (i) reinforcement-based learning (RL) through trial and error, and (ii) belief-based learning through anticipating and responding to actions of others[18]. In particular, RL models posit that learning is driven by a prediction error defined as the difference between expected and received rewards and have been highly successful in connecting behavior to the underlying neurobiology[28,29]. In contrast, belief-based learning posits that players make use of knowledge of the structure of the game to update value estimates of available actions and comes in two computationally equivalent interpretations. One interpretation assumes the existence of latent beliefs and requires players to form and update first-order beliefs regarding the likelihood of future actions of opponents. Specifically, under this interpretation, the model posits that players select actions strategically by best responding to their beliefs about future strategies of opponents and update these beliefs by using some weighted history of opponents' choices[18,30]. Mathematically, players engaging in belief learning correspond to Bayesian learners who believe opponent's play is drawn from a fixed but unknown distribution and whose prior beliefs take the form of a Dirichlet distribution[18]. Under the alternative interpretation, beliefs and mental models are not assumed and action values are updated directly by reinforcing all actions proportional to their foregone (or fictive) rewards[31]. The equivalence of these two mathematical interpretations thus makes it clear that belief-based learning does not necessarily imply the learning of mental, verbalizable beliefs commonly referred to in the cognitive and social sciences, because specific beliefs about likely strategies of opponents are sufficient but not necessary for this type of learning.

Importantly, previous neuroimaging results have been able to disaggregate distinct computational signatures of reinforcement-based learning (RL) and belief learning processes based on trial-by-trial variation in neural responses along frontostriatal circuits. Specifically, whereas the medial prefrontal cortex (mPFC) selectively responds to belief learning prediction error signals, activity in the putamen, a substructure of the BG, is correlated with prediction errors associated with both RL and belief-based learning[7]. Building on these findings, therefore, we sought to investigate the extent to which putative computational processes such as RL and belief-based learning would reflect functions of BG necessary for social and strategic learning.

To this end, we studied behavior of focal lesion patients with damage in either BG ($N = 6$) or OFC ($N = 6$), and a cohort of healthy comparison (HC) subjects ($N = 20$; Fig. 1a; Methods; Supplementary Table 1), which is consistent with sample sizes used in previous lesion studies in the field of cognitive neuroscience[10,32–35]. Following informed consent, subjects were tested in the Patent Race as well as a matching non-strategic version where we replaced the human pool players with a computer algorithm. Specifically, in the non-strategic task, participants were told to choose an investment that must exceed a randomly generated hurdle to succeed, but were not told how the computer generated the random hurdle choices (Methods). Importantly, whereas in strategic learning both belief and reinforcement components were engaged, our previous work has shown that learning in the non-strategic environment was driven primarily by reinforcement learning[7]. Thus the inclusion of both strategic and non-strategic conditions makes opposing predictions about how BG lesion will affect learning in the economic game. If BG are involved in both trial-and-error learning as well as social functioning such as strategic learning, damage to BG will affect performances in both strategic and non-strategic environments. Alternatively, if, in the strategic environment where multiple learning processes are engaged, learning inputs originated from prefrontal areas provide compensatory functions for trial-and-error deficits resulted from the BG damage, we should expect that damage to the BG selectively impairs learning capacity in the non-strategic, reward-reinforcing environment, as opposed to the more complex, interpersonal, strategic environment.

Consistent with neuroimaging evidence suggesting dissociable contributions of the BG and prefrontal cortex (PFC) to multiple learning rules, we found that patients with BG damage performed similarly as participants in the HC cohort in the strategic condition, where learning was driven by a mixture of reinforcement and belief learning. In contrast, BG patients were markedly impaired when information about the strategic context was removed, such that participants must rely primarily on learning based on reinforcement. Moreover, these differences were qualitatively distinct from those observed in patients with OFC damage, suggesting that findings related to BG cannot be attributed to the specific method used in the study or a general deficit associated with reward circuit damage.

## Results

**Overall task performance**. To characterize overall task performance and differences across cohorts in the Patent Race game (Fig. 1b), we first examined the extent to which participants' likelihood of changing their choices on a round-by-round basis was affected by received and foregone payoff in each round (Fig. 2a). In particular, this analysis captures the idea from previous theoretical and empirical studies showing that whereas behavior of pure reinforcement learners should be sensitive only to received payoffs, belief-based learners will be in addition sensitive to foregone payoffs.

To illustrate this, suppose that the Weak player observes the Strong players frequently investing five units. She may subsequently respond by playing zero to keep her initial endowment. Upon observing this play, Strong players can exploit the Weak player's behavior by investing only one unit to obtain the prize while keeping four units from the endowment. This behavior may, in turn, entice the Weak player to move away from investing zero to win the prize. In contrast, pure RL players will respond to these changes in behavior of the opponents in a much slower manner, because they behave by comparing received payoffs from past investments without consideration for the strategic behavior of others (Supplementary Figure 1).

We therefore conducted model-free logistic regression of the probability that HC participants would choose the same strategy on the received payoff and foregone payoff on a round-by-round basis. As there were multiple foregone payoffs in a round, we operationalized foregone payoffs by taking the difference between maximal foregone payoff and received payoff, which following previous literature, we refer to as "regret" (Fig. 2a)[18,36]. Consistent with previous theoretical and empirical findings, we found that the

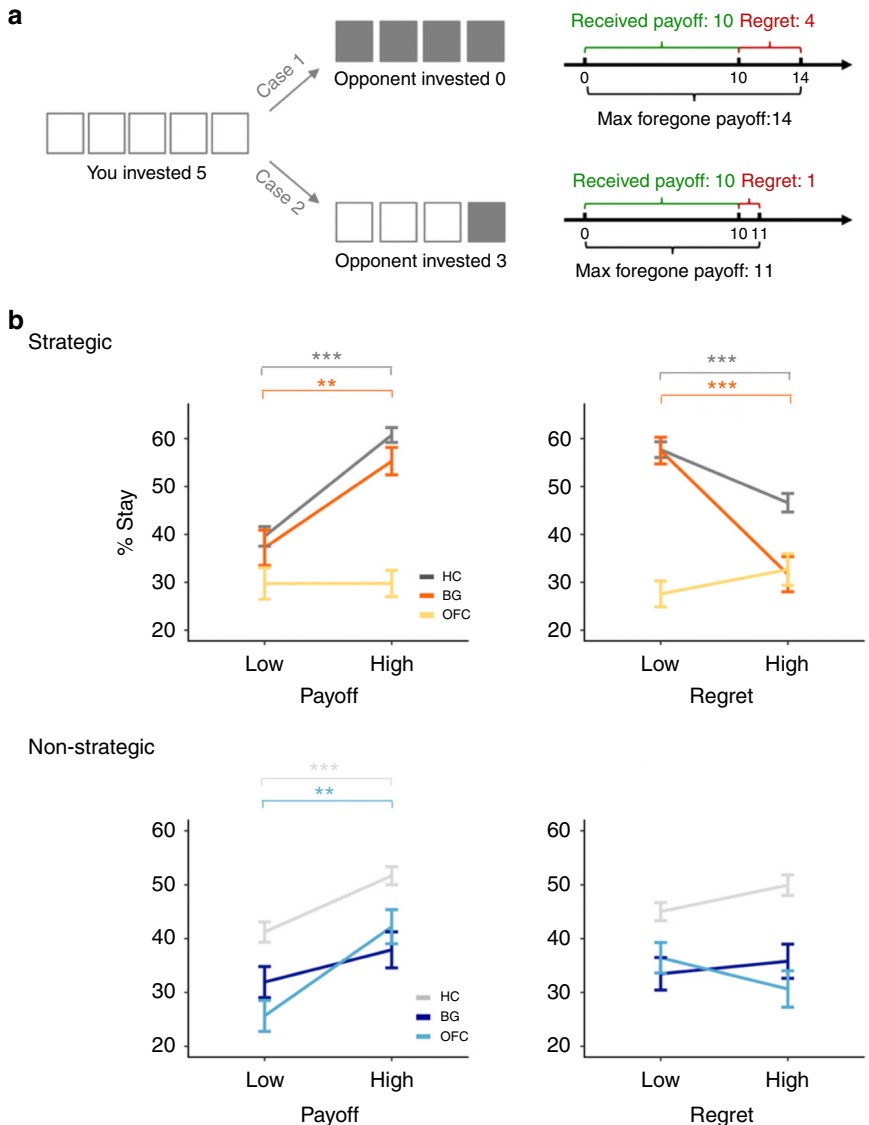

**Fig. 2** Overall task performances. **a** Illustration for the received and forgone payoff. The received payoff reflects the amount of reward obtained through the action chosen by the subject, whereas the maximal forgone payoff reflects the amount of reward one could have received by choosing the best available action given the opponent's decision. For example, if the subject chose 5 and the opponent 0, the maximal forgone payoff was 14, as the optimal action would have been to invest 1, as opposed to the received payoff 10. Had the opponent chosen 3, however, the received payoff would remain to be 10 but the maximal payoff would be 11, as choosing 4 would have been the best strategy. Thus, the maximal forgone payoff reflects the variability in opponents' actions and their effects on possible outcomes. Following previous studies, regret is defined as the distance between the maximal forgone and received payoff, given the opponent's decision on a particular trial. **b** Stay/switch frequency versus received payoff (left) or regret (right). Y-axes represent the percentage of trials in which subjects chose to stay with the same decision on the next trial (i.e., "stay"), based on median splits on payoffs and regrets for each cohort and condition (x-axes). Error bars represent S.E.M. $^{*}P < 0.05$, $^{**}P < 0.01$, $^{***}P < 0.001$, Bonferroni corrected

extent to which HC participants would stay (switch) with the same strategy was associated with having received high (low) payoff or low (high) regret. Specifically, the probability that HC participants repeated the same choice in round $t$ as in $t−1$ was significantly associated with the size of the payoff ($\beta = 0.86$, Bonferroni 95% confidence interval $= (0.56,1.17)$, $P < 0.001$, Bonferroni corrected; all reported $p$ values are two-tailed), and negatively associated with the size of regret ($\beta = −0.45$ ($−0.74,−0.15$), $P < 0.001$, Bonferroni corrected; Fig. 2b, Table 1).

Moreover, to manipulate the relative contribution of these two learning rules to behavior, we altered the social context by testing subjects in a matching "non-strategic" condition, where we replaced human pool opponents in the Patent Race with a matching computer algorithm (Methods). Previous work has

shown that, whereas participants respond to both received and foregone payoffs in a strategic environment, in the non-strategic environment learning is driven primarily by reinforcing actions associated with the received payoff[18,36]. In line with previous findings, HC participants in the non-strategic condition showed significant sensitivity to the received payoff but not regret ($\beta = 0.42$ ($0.13,0.72$), $P < 0.001$, and $\beta = 0.20$ ($−0.09,0.49$), $P > 0.05$, respectively, Bonferroni corrected), such that actions that were recently rewarded, regardless of the level of regret, were more likely to be repeated in the subsequent round (Fig. 2b, Table 1).

Using these measures, we next investigated how lesion to BG or OFC affected performance across strategic and non-strategic conditions. We found that, similar to HC participants, BG patients displayed significant sensitivity to both payoff and regret

**Table 1 Logistic regression modeling of effect of received payoff and regret on subsequent stay/switch decisions**

| | Strategic | | | Non-strategic | | |
| | HC | BG | OFC | HC | BG | OFC |
|---|---|---|---|---|---|---|
| Payoff | 0.86*** | 0.74** | 0.00 | 0.42*** | 0.26 | 0.75** |
| | (0.56, 1.17) | (0.18, 1.30) | (−0.58, 0.59) | (0.13, 0.72) | (−0.29, 0.82) | (0.19, 1.33) |
| Regret | −0.45*** | −1.07*** | 0.24 | 0.20 | 0.10 | −0.26 |
| | (−0.74, −0.15) | (−1.67, −0.49) | (−0.34, 0.82) | (−0.09, 0.49) | (−0.45, 0.66) | (−0.84, 0.31) |

Significantly positive coefficient for payoff suggests increased likelihood to stay as the received payoff increases, and negative coefficient for regret suggests increased likelihood to stay as the regret decreases. *$P < 0.05$, **$P < 0.01$, ***$P < 0.001$, Bonferroni corrected. Parentheses contain Bonferroni-corrected 95% confidence interval

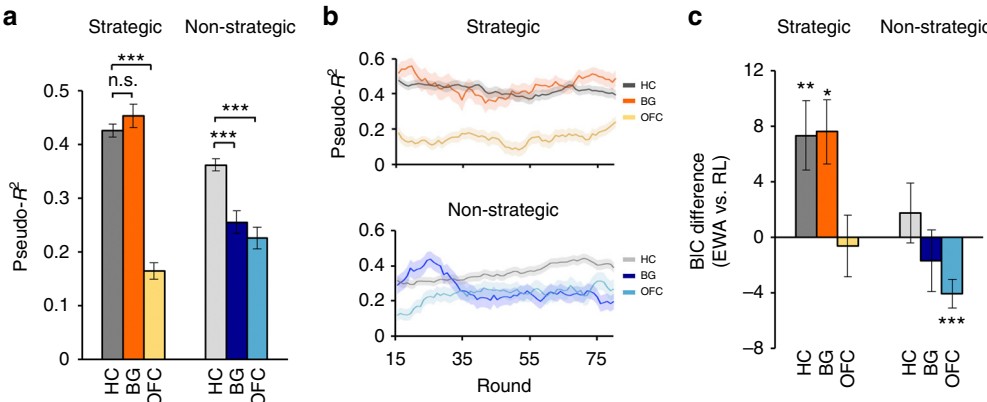

**Fig. 3** Computational modeling. **a** Differential ability of EWA in explaining choice behavior. The bar plots depict values of pseudo-$R^2$ derived from the best-fitting EWA, defined as the difference between the log-likelihood of the EWA model and a random choice model, scaled by log-likelihood of the random model. Higher pseudo-$R^2$ values indicate better model fit relative to chance level. The means and error bars were constructed using a bootstrap procedure with 10,000 iterations pooling over cohorts for each condition. **b** Trial-level EWA model fit based on pseudo-$R^2$ plotted using 15-trial bins. **c** Bayesian Information Criterion (BIC) showing significant improvement of the hybrid model fit relative to the baseline RL model, in HC and BG cohorts under the strategic treatments, calculated using a bootstrap sampling procedure with 10,000 iterations. Error bars indicated bootstrap S.D. and shaded areas indicate S.E.M. *$P < 0.05$, **$P < 0.01$, ***$P < 0.001$, Bonferroni corrected

in the strategic condition ($\beta = 0.74$ (0.18,1.30), $P < 0.01$, and $\beta = -1.07$ (−1.67,−0.49), $P < 0.001$, respectively, Bonferroni corrected; Fig. 2b, Table 1). In stark contrast, in the non-strategic condition where the information about the strategic context was removed, BG patients were not sensitive to either the payoff or regret ($\beta = 0.26$ (−0.29, 0.82), $P > 0.05$, and $\beta = 0.10$ (−0.45, 0.66), $P > 0.05$, respectively, Bonferroni corrected; Fig. 2b, Table 1). Interestingly, decisions of the OFC cohort exhibited the opposite pattern, displaying little responsiveness to either payoff or regret in the strategic condition ($\beta = 0.00$ (−0.58, 0.59), $P > 0.05$, and $\beta = 0.24$ (−0.34, 0.82), $P > 0.05$, respectively, Bonferroni corrected), but significant sensitivity to the received payoff in the non-strategic condition ($\beta = 0.75$ (0.19, 1.33), $P < 0.01$, Bonferroni corrected; Fig. 2b, Table 1). All results were robust to analyses controlling for demographic variables and neuropsychological assessments, as well as non-parametric permutation tests sampling null distribution for each cohort and each condition (Supplementary Table 2 and Supplementary Figure 2). In addition, as the stay/switch measure does not take into account the potential difference in switching to more or less-adaptive strategies, additional analyses were performed to examine the frequency of choosing optimal actions by each cohort under each condition, based on model-free measures of optimal choices (Supplementary Figure 3).

**Intact strategic learning capacity following BG damage**. The above results therefore argue in favor of a model where effects of the BG damage on social functioning were buffered by other

learning processes, but not when the social context was removed. To more formally test this dissociation, and to connect behavioral differences to underlying cognitive mechanisms, we applied a computational approach using the Experience-Weighted Attraction (EWA) model, which nests reinforcement and belief-based learning algorithms as special cases and has been highly successful in connecting these computational components with neural responses along frontostriatal circuits (Supplementary Figure 1; see Methods)[7,31,37].

We tested the hypothesis that the extent to which BG are asymmetrically involved in learning in strategic and non-strategic environments would be reflected by the differential ability of EWA to explain choice behavior. That is, BG patients should benefit more when shifting from the non-strategic to strategic environment, in terms of the EWA model fit either in-sample (e.g. pseudo-$R^2$) or out-of-sample (e.g., hold-out prediction accuracy), compared with that of HC subjects. Specifically, comparing pseudo-$R^2$ values[31], we found no significant difference in how well EWA explained choice behavior under the strategic condition in BG vs. HC cohorts (BG: mean±standard error of mean = 0.45 ± 0.02; HC: 0.42 ± 0.01; bootstrapped 95% CI = (−0.03, 0.09), $P > 0.05$, Bonferroni corrected; Fig. 3a; Methods). In contrast, in the non-strategic condition, the BG cohort was associated with significantly lower pseudo-$R^2$ than that of HC (BG: 0.26 ± 0.02; HC: 0.36 ± 0.01; bootstrapped 95% CI = (−0.17, −0.05), $P < 0.001$, Bonferroni corrected; Fig. 3b).

Moreover, there was a significant cohort (BG vs. HC) by condition (strategic vs. non-strategic) interaction, such that BG

damage was associated with a more pronounced increase in EWA model fit from the non-strategic to strategic condition compared with that of HC participants (increase of model fit in BG: $0.20 \pm 0.03$; HC: $0.06 \pm 0.02$; bootstrapped 95% CI = (0.05, 0.22), $P < 0.001$, Bonferroni corrected). Similar results were obtained when comparing the EWA explanatory power between BG and HC at either trial- or subject-level (Fig. 3b, Supplementary Figure 4–5). To address potential concerns regarding overfitting and spurious cohort differences arising from natural variations in learning across individuals, we performed additional analyses using out-of-sample tests and permutation tests shuffling cohort labels. Both yielded similar results (Supplementary Figure 6–7).

In contrast, the behavior of OFC patients was associated with significantly lower explainable variances than HC participants in both strategic (OFC: $0.16 \pm 0.02$ vs. HC: $0.42 \pm 0.01$; bootstrapped 95% CI = $(-0.31, -0.21)$, $P < 0.001$, Bonferroni corrected) and non-strategic conditions (OFC: $0.23 \pm 0.02$ vs. HC: $0.36 \pm 0.01$; bootstrapped 95% CI = $(-0.19, -0.08)$, $P < 0.001$, Bonferroni corrected; Fig. 3a). Interestingly, there was also some evidence for a significant cohort by condition interaction, such that the OFC damage was associated with a decrease in pseudo-$R^2$ from the non-strategic to strategic condition comparing with healthy participants (increase of model fit in OFC: $-0.06 \pm 0.02$; HC: $0.06 \pm 0.02$; bootstrapped 95% CI = $(-0.20, -0.05)$, $P < 0.001$, Bonferroni corrected). However, unlike in BG patients, OFC effects were sensitive to alternative specifications such as the self-tuning estimation, where some of the EWA parameters were replaced by functions of experience of OFC patients[37] (Supplementary Figure 8). Model estimates and additional robustness checks are reported in the Supplement (Supplementary Figure 9, Supplementary Tables 3–4).

**Compensatory role of higher-order learning inputs**. To more formally test the hypothesis that BG damage spares the capacity to engage in belief-based learning, we used the EWA model to disentangle the relative contributions of different decision rules across cohorts in strategic and non-strategic conditions. Specifically, we examined the extent to which EWA improved the explanatory power above and beyond the basic RL algorithm. That is, if strategic learning capacity in BG patients was compensated using high-order learning processes, EWA should significantly improve the fit relative to the baseline RL (Supplementary Figure 1). By focusing on model comparison as opposed to specific parameters calibrated from the behavior (e.g., the weight on belief-based learning), this method is less dependent upon the accurate identification of model parameters, which can be problematic particularly in lesion cohorts associated with poor model fits. Importantly, this test also serves as a more stringent test, because choices that were equally explainable by RL and other learning rules nested within EWA were attributed solely to the RL algorithm.

Using the Bayesian Information Criterion (BIC) to penalize for the number of free parameters, we found that in control subjects, consistent with previous studies, EWA significantly improved the fit only in the strategic but not the non-strategic condition (Strategic: $7.36 \pm 2.50$, bootstrapped 95% CI = (1.71, 14.74), $P < 0.01$; non-strategic: $1.76 \pm 2.16$, bootstrapped 95% CI = $(-3.51, 7.70)$, $P > 0.05$, Bonferroni corrected; Fig. 3c). Critically, in the strategic condition, EWA significantly improved the fit of BG patients relative to the baseline RL ($7.62 \pm 2.29$, bootstrapped 95% CI = (1.28, 12.63), $P < 0.05$, Bonferroni corrected), but not in the non-strategic condition ($-1.68 \pm 2.24$, bootstrapped 95% CI = $(-7.11, 4.27)$, $P > 0.05$, Bonferroni corrected). Finally, in the OFC cohort, EWA did not explain the choice behavior above and beyond the basic RL model in the

strategic condition ($-0.61 \pm 2.21$, bootstrapped 95% CI = $(-5.14, 5.75)$, $P > 0.05$, Bonferroni corrected), and in fact was significantly worse than RL in non-strategic condition after penalizing for additional parameters ($-4.06 \pm 1.04$, bootstrapped 95% CI = $(-6.37, -1.15)$, $P < 0.001$, Bonferroni corrected) (Fig. 3c, Supplementary Figure 5).

## Discussion

A wealth of neuroimaging data has implicated the involvement of the BG, and in particular the striatum, in a striking variety of goal-directed decisions, including those involving acquiring rewards for oneself as well as in the social domain where actions and outcomes depend on rewards of others[4,7,24,38]. In the former, these correlational findings have been corroborated with findings from causal studies using focal lesion patients and those with neurodegenerative disorders known to affect BG[11,19,22–24,39–41]. In contrast, surprisingly little evidence exists, either in support of or argue against, the causal involvement of BG in social decision making[11,13,16].

Here by connecting the lesion method with neuroeconomic tools, we show that capacity for strategic learning in the presence of competitive, intelligent opponents can be preserved in patients with focal BG damage, despite having deficits in learning in a non-social, probabilistic environment. Model comparisons further show that damage to BG spares strategic learning capacity possibly through compensatory processes such as belief-based learning when the social context is available for anticipating future actions of others.

Owing to variation in lesion location and extent across patients, it is possible that our findings were driven by damage to specific BG nuclei or adjacent regions outside of BG. The maximal lesion overlap in our sample of BG patients is in the ventral rostral putamen (6/6), as well as in the globus pallidus (4/6) and caudate (2/6) (Fig. 1a). In particular, the putamen and caudate nucleus have been previously implicated in learning about actions and their reward consequences in action-contingent learning[42–45] and in social exchanges involving trust and reputation that requires learning about social agents based on their previous actions[4]. Across analyses, however, there is no association of performance with lesion extent or their location along the dorsal/ventral axis, and findings are robust to exclusion of patients with caudate lesions (Supplementary Tables 5–6). Similarly, damage extending to the insular cortex, which was observed in three of six patients in the BG lesion cohort, was not associated with performance (Supplementary Table 5). In contrast, behavior in patients with damage to the OFC, another critical node within the reward circuit, shows a qualitatively distinct pattern, suggesting that findings related to BG cannot be attributed to the specific method used in the study or considered as a general property associated with the reward circuit.

Together with prior neuroimaging findings, our data provide insights into the computational underpinnings of social decision making and the apparently contradictory findings from past neuroimaging and lesion studies. Specifically, both set of findings are consistent with a model of BG functioning in receiving higher-order learning signals broadcasted from other regions involved in social cognition to the striatal input areas. In line with this model, BG activations identified in prior fMRI studies of social decision-making were typically accompanied by concurrent activations in other brain regions involved in social cognitive processing, including the rostral anterior cingulate cortex (ACC)[46], mPFC[7,47], and temporoparietal junction (TPJ)[48,49]. Within the Patent Race itself, BOLD responses in the putamen were found to be associated with prediction errors arising from both belief-based and reinforcement learning, whereas activity in

the medial PFC is correlated with belief-based learning prediction errors[7].

Our results are consistent with past studies in BG disorders, suggesting the presence of compensatory processes when the task can be solved through multiple learning strategies. For example, although BG damage is associated with impaired learning in changing, probabilistic environments[19,23,39], there is some evidence that learning capacity is intact when patients are able to engage in declarative learning strategies which do not depend on the integrity of BG[39]. Results of these studies thus raise intriguing questions regarding whether the asymmetrical functions of BG are specific to strategic vs. non-strategic comparison, or hold in more general settings of social decision-making where multiple cognitive processes are supported by dissociable neural systems. For example, in observational learning where individuals can learn from either the actions or choice outcomes of others in a non-strategic manner, it remains unclear whether the putative contribution of BG in the processing of outcome-based learning signals is necessary for learning from observations, or can be compensated by action-based learning that depends on the dorsolateral PFC[6].

An alternative possible explanation is that the preserved strategic learning capacity reflects the compensatory role of the intact contralateral BG. Indeed, owing to the rarity and often devastating motor deficits of bilateral BG damage[11], our BG cohort consisted only of those with unilateral lesion. As a result, it is possible that the intact hemisphere alone is sufficient for learning in social settings, but not in non-social environment. More broadly, it suggests the possibility where social learning capacity following BG damage may crucially depend on intact functional coupling between preserved portions of the BG and cortical regions involved in social cognitive processes. This is consistent with existing causal evidence from both lesion and TMS studies, demonstrating the causal involvement of cortical regions, including the ACC[50–52], mPFC[10,51,53], and right TPJ[54,55], in social decision-making in humans and non-human primates. Future studies comparing the functional connectivity of these regions in patient vs. healthy populations would be valuable in understanding how socially relevant information are integrated during such decision processes[3–5].

Interestingly, although pseudo-$R^2$ values were fairly consistent over time in the strategic condition, they were more variable in the non-strategic condition. This is particularly true for the BG patients during rounds 15–35, where pseudo-$R^2$ dropped sharply following a rise at the start of the experiment. This may reflect the engagement, albeit less successfully than in the strategic condition, of compensatory mechanisms in BG patients in the non-strategic condition, for example, through relying on working memory (WM) systems. Indeed, past studies of reward learning suggest that prediction errors produced by RL systems include significant contribution from WM systems, especially during early learning[56,57]. Future studies are needed to more firmly establish this effect and the underlying neural mechanisms.

Our findings also contribute to the understanding of social deficits associated with OFC lesions. Deficits observed in our OFC patients were particularly marked in the strategic condition, hinting at a more pronounced impairment during decision-making in social contexts. This is consistent with the wealth of neuropsychological findings documenting profound changes in social behavior following the OFC damage, including impaired capability in perspective taking and inferring mental states of others[10]. The OFC effects observed in our experiment, however, were more heterogeneous and sensitive to the specific analytic choices. One candidate explanation is that this reflects the greater variation in the damage extent in our OFC cohort, which in some cases extended into the lateral and dorsal regions. Previous literature suggests lateral and medial OFC differentially contribute to processes entailing, respectively, learning and updating versus those involved in value comparisons especially in decisions among three or more options[58–60]. Moreover, owing to the presence of white matter damage and in some cases adjacent regions including the lateral OFC, we cannot completely rule out the contribution from non-mOFC based processes[61]. Future experiments with larger sample sizes in combination with lesion mapping techniques will be needed to test these possibilities.

A more general concern with our model-based approach is the possibility of model misspecification due to participants engaging in decision rules beyond the EWA model space[62]. This is particularly the case with lesion cohort behavior, as model-based approaches such as RL or EWA inherently make strong assumptions regarding how past experiences are integrated over the course of learning. Our study addressed this in two ways. First, we focused on cross-cohort comparisons using goodness-of-fit measures, rather than specific parameters calibrated from behavior (e.g., the weight on belief-based learning). In particular, comparisons based on individual parameters provide meaningful insights into cognitive components if cohorts behave in accordance to model assumptions. For example, using comparison between HC strategic and non-strategic conditions, the belief learning parameter provided a good indication that HC relied less on belief learning in the non-strategic condition. On the other hand, a poor model fit raises the possibility that one or more model assumptions are violated. This misspecification issue is equally true for both frequentist and Bayesian approaches. More importantly, such model misspecification can result in either upward or downward biases in parameter estimates. This makes it difficult, even in the presence of significant differences in parameter estimates, to draw firm conclusions regarding differences in cognitive mechanisms between cohorts or conditions.

Second, consistent with other neuroeconomic studies using the lesion method (e.g.,[33,34,58,63,64]), we used model-free analyses whenever possible to characterize behavioral deficits and support conclusions derived from model-based methods. Specifically, this involved examining the extent to which participants' choice behavior was sensitive to various different notions of learning signals without restricting the specific functional form that weights on these signals may take (Supplementary Figure 10). A more thorough investigation using data-driven approaches will be needed to further assess and compare choice predictability of lesion patients vs. control subjects, removing assumptions regarding which and how external stimuli drive the learning process.

Issues of whether, and under what circumstances, cognitive processes supported by BG reflect the computational properties necessary for social behavior have important implications for understanding the interaction between parallel cognitive processes, as well as the neural mechanisms necessary for arbitrating between such processes. The present study thus demonstrates the utility of combining the lesion method with formal models of behavior in addressing these questions. At the same time, an important limitation of our study concerns the limited sample size of patient cohorts, particularly given the inherent rarity of focal BG lesion. Future studies can address this issue by using lesion analytical methods, such as model-based lesion symptom mapping, to identify the distributed patterns of brain areas within BG that subserve social functions. In addition, future studies also need to address whether our findings generalize to other types of social decisions, including those involving prosocial motivations[47,65,66], reciprocity[67], and social dominance[53].

## Methods

**Subjects**. Patients with focal brain lesions to the BG ($n = 6$) and OFC ($n = 6$) were included in the experiment. HC participants ($n = 20$) were recruited from San Francisco Bay Area, CA. All subjects provided informed consent and the study was approved by the Committee for Protection of Human Subjects at the University of California, Berkeley, CA. See Supplementary Table 1 for demographic information and neuropsychological background of lesion patients.

**Lesion reconstruction**. Software reconstructions were performed using MRI-cron[68]. For both patient groups, testing took place at least 6 months after the date of the stroke/accident. A neurologist (R.T.K.) inspected patient MRIs to ensure that no white matter hyperintensities outside the lesioned area were observed in either patient group. All TBI patients had low impact force injuries with no clinical or MRI evidence of axonal shear.

**Procedure**. Following task instructions and a comprehension quiz, participants were administered two blocks of strategic and non-strategic condition trials, each containing 80 rounds. All choices were conducted using hypothetical payoffs and no feedback, with order of the strategic and non-strategic blocks counterbalanced across participants within each cohort. We first conducted a behavioral session where 16 healthy subjects played the Patent Race for two sessions of 80 rounds each using a random matching protocol. We will refer to these subjects as "pool players". Players switched Strong and Weak roles at the end of the first 80 rounds.

In the strategic condition, lesion patients and healthy control subjects played against these pool players. Importantly, they played in the same sequence as pool players. For example, if the patient was on round 60, the opponent's choice would be drawn randomly from round 60 of one of the pool players. We used the random matching protocol in the strategic condition for two reasons. First, this protocol requires a reasonable number of subjects to ensure that the probability of repeated interaction is small. Otherwise, subjects may be able to develop hierarchical mental models in order to collude with or trick the paired opponents. Second, it helps to preserve the dynamics of the evolution of play in the experiment, and control for the inter-group variation that would arise if we used more than one group of pool subjects.

In the non-strategic condition, we replaced the human pool players with a matching computer algorithm. Participants were told to choose an investment that must exceed a randomly generated hurdle to succeed, and were not told how the computer generated the hurdle. In behavioral pilots, we generated randomness using two different methods. The first is a stationary distribution using the proportion of strategies chosen in the human population. The second method essentially replicates the human treatment, in that we sampled trial-by-trial from the empirical time series of the population play from the human pool players. We found that learning in both non-strategic treatments was driven by reinforcement rather than belief learning. Here we used the second method to precisely match the sequence of stimuli across social and non-social conditions.

**Computational modeling**. To quantitatively characterize relative contribution of reinforcement and belief-based inputs to behavior, we used the well-established hybrid model of experience-weighted attraction (EWA) first introduced by Camerer and Ho[31]. According to this model, player $i$ updates the expected value of each strategy $k$ at trial $t$ (denoted by $V_i^k(t)$) according to the following rule:

$$V_i^k(t) = \begin{cases} \frac{\phi \cdot N(t-1) \cdot V_i^k(t-1) + \pi_i\left(s_i^k, s_{-i}(t)\right)}{N(t)}, & \text{if } s_i^k = s_i(t), \\ \frac{\phi \cdot N(t-1) \cdot V_i^k(t-1) + \delta \cdot \pi_i\left(s_i^k, s_{-i}(t)\right)}{N(t)}, & \text{if } s_i^k \neq s_i(t), \end{cases} \tag{1}$$

where $N(t) = \rho \cdot N(t-1) + 1$.

Here, player $i$'s strategy of investing $k$ at trial $t$ is denoted as $s_i^k(t)$, and the strategy chosen by $i$'s opponent is denoted as $s_{-i}(t)$. Variable $\pi_i(s_i^k, s_{-i}(t))$ thus represents the payoff player $i$ will receive if he/she invests $k$ in trial $t$, given his/her opponent's choice $s_{-i}(t)$. The key insight of the EWA is that it allows a player updating both the expected value of the chosen option based on the received payoff, and the value of unchosen options based on foregone payoffs given the opponent's decision at a particular trial. The former corresponds to the standard RL algorithm, and the latter is equivalent to the belief-based learning rule. The parameter $\delta$ controls the extent to which the player weights foregone payoffs relative to the received payoff. Parameter $\phi$ is a discount factor that depreciates previous subjective values. Function $N(t)$ captures the importance of the pre-game experience in updating the expected value of a choice option based on experiences gained within the game. Parameter $\rho$ controls how fast $N(t)$ decays. When $\delta = 0$, $N(0) = 1$, and $\rho = 0$, the EWA model reduces to a basic RL model:

$$V_i^k(t) = \begin{cases} \phi \cdot V_i^k(t-1) + \pi_i(s_i^k, s_{-i}(t)), & \text{if } s_i^k = s_i(t); \\ \phi \cdot V_i^k(t-1), & \text{if } s_i^k \neq s_i(t). \end{cases} \tag{2}$$

**Behavioral data analysis**. To calibrate EWA and RL models given subjects' behavior in the experiment, we estimated the parameters of each model by maximizing the log-likelihood of model predictions for each cohort and each condition separately. Expected values were then converted into choice probability through the well-established softmax function:

$$p_i^k(t+1) = \frac{e^{\lambda \cdot V_i^k(t)}}{\sum_{l=1}^L e^{\lambda \cdot V_i^l(t)}}, \tag{3}$$

where $p_i^k(t+1)$ is the probability of investing $k$ at trial $t+1$ for player $i$, and $\lambda$ is the inverse temperature. Based on these choice probabilities, we maximized $\sum_i \sum_t \log\left(p_i^{s_i(t)}(t)\right)$, the sum of log probability of each subject's observed choice at each trial. Different combinations of initial values of key parameters were used for searching, as the likelihood function might not be globally concave.

Pseudo-$R^2$ was derived based on the best-fitting EWA model as an indicator of model performance. It is defined as the difference between the log-likelihood of EWA for the given cohort and the log-likelihood of a random choice model, scaled by log-likelihood of the random model[31]. The higher the pseudo-$R^2$, the better EWA in predicting decisions over the chance level. The means and error bars of the pseudo-$R^2$ were constructed using a bootstrap procedure pooling over cohorts for each condition. Specifically, on each iteration, we constructed a pseudo-sample of a specific cohort and condition by resampling, with replacement, the pseudo-$R^2$ of all trials across all subjects within that cohort. To account for the potential issue of EWA model overfitting, we also performed an out-of-sample prediction validation by estimating the hybrid EWA model using the first 60 trials of each subject and tested on the last 20 trials to obtain the prediction accuracy[31,48].

BIC was calculated by pooling over participants in each cohort, and dividing by the sample size of the cohort to account for differences in number of participants in each cohort. BIC error bars were calculated using a bootstrap procedure pooling over cohorts across conditions. Specifically, on each iteration, we constructed a pseudo-sample of a specific cohort and condition by resampling, with replacement, the log-likelihood values of all trials across all subjects within that cohort, and penalizing the additional number of parameters to derive the BIC.

**Power analyses**. Our choice of sample size was guided by previous lesion studies in the field of cognitive neuroscience[31-35], supplemented by power calculations that evaluated the sensitivity of our design using a model-based simulation approach. Specifically, we evaluated the power of detecting a difference in pseudo-$R^2$ of EWA model fit between HC and lesion patients who were assumed to behave randomly. Because our pilot HC data indicated that values of pseudo-$R^2$ for the best-fitting EWA were 0.43 and 0.45 in the strategic and non-strategic condition, respectively, with standard deviation of 0.2 for both, we conducted our power analysis using the same mean estimates for HC and 0 (pseudo-$R^2$ for random choice model) for patients, assuming the same estimates of 0.2 as standard deviation for both cohorts. Our computation, under alpha level 0.05 and power $(1 - \beta) = 0.8$, indicated a sample size of four patients is adequate to detect the difference in model fit where they exist.

**Reporting summary**. Further information on experimental design is available in the Nature Research Reporting Summary linked to this article.

## Data availability

The data and code that support the findings in this article are available at Open Science Framework: https://osf.io/4x3nf/.

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

## Acknowledgements

We thank D. Walsh and C. Clayworth for assistance with data collection, analyses, and lesion reconstruction. This research was supported by NSFC (31671171 and 31630034 to L.Z.) and the National Institutes of Health (MH098023 and DA043196 to M.H.).

## Author contributions

L.Z. and M.H. designed research; L.Z. and D.S. performed research; L.Z., Y.J., R.T.K. and M.H. carried out statistical analyses; and all authors wrote the paper.

## Additional information

**Competing interests:** The authors declare no competing interests..

**Journal Peer Review Information**: *Nature Communications* thanks the anonymous reviewer(s) for their contribution to the peer review of this work. Peer reviewer reports are available.

