## [Peer Review File · Nature Communications]

Reviewers' comments:

Reviewer #1 (Remarks to the Author):

Zhu and colleagues studied decision-making in three populations of individuals, patients with basal ganglia (BG) lesions, patients with orbitofrontal cortex (OFC) lesions, and healthy controls. Their central aim was to determine why patients with OFC lesions but not patients with BG lesions often display deficits in social decision-making, when many of the known BG functions should be also important for underlying mechanisms of decision-making. Therefore, this is a highly interest line of inquiry with a broad interest for the field of decision neuroscience.

In order to tackle this question, the authors employed strategic and non-strategic versions of a competitive game known as the Patent Race. While the strategic version of this game involved a mix of both belief-based and reinforcement-based learning, the non-strategic version involved reinforcement-based learning more exclusively. While OFC, but not BG, patients displayed deficits relative to healthy controls in the strategic version of the task, BG patients displayed deficits only in the non-strategic version of the task, suggesting that patients with BG lesions have a dissociable deficit in reinforcement-based learning that can be potentially compensated in the strategic context by intact belief-based learning. This is an interesting and important finding that helps toward clarifying why patients with BG lesions often do not show social decision deficits despite the fact that BG is involved fundamental aspects of learning and decision-making in general. The finding is robust, including nice analyses of behavior from a unique set of patient populations combined with sophisticated modeling. I only have a couple of comments with a goal of further improving the manuscript and clarifying less-explained results. In sum, this is an exciting finding and also a broadly important finding on how BG and OFC contributes to decision-making.

1. It is clear that the conceptual framework provided for the interpretation of BG lesions is interesting and is in nice correspondence with the results observed. The results concerning OFC lesions, however, warrant some more clarifying explanation from the authors under the same framework. The authors suggest that in order to perform well in the strategic version of their task, people need either belief or reinforcement-based learning to be intact. For the non-strategic version of their task, only reinforcement-based learning is relevant, so only this needs to be intact for performance to be adequate. That logic fits the BG lesion results very well - BG patients have trouble on the task that only requires reinforcement-based learning, but can perform adequately on the strategic version of the task because their belief-based learning is intact. But, according to this framework (i.e., OFC may be involved in both belief and reinforcement-based processes), it is plausible that the OFC patients may also have some trouble in the non-strategic task that only requires the reinforcement-based system. However, at the behavioral level, the OFC lesion group showed intact (and strong) low-high payoff relationship in the non-strategic task (Fig. 2b,c). On the other hand, the section in the Results outlining the Experience-Weighted Attraction (EWA) model found that the OFC lesions led to similarly low pseudo-R2 as the BG group (Fig. 3a) (although Fig. S5 results become much closer to Fig. 2b's data). Some effort was made to explain these findings at the end of the paragraph starting on line 215, but more explanation could help clarify and consolidate these findings. Basically, I think the authors may want to add more explanation so that consolidating across the behavioral and modelling results become clearer for the readers.

2. In Fig. 3b, time courses are shown for the pseudo-R2 value for consecutive trials. While pseudo-R2 is fairly consistent over time in the strategic task, this value seems to change noticeably over time in the non-strategic task, especially for the BG patients (rounds 15-35 vs. the rest where it stabilizes). Please provide further explanation either in the results or discussion sections since it suggests some shift in the BG patients' behaviors occurring relatively early in the game.

Reviewer #2 (Remarks to the Author):

In this paper, Zhu et al. administered an economic game, called the Patent Race, which they have previously investigated in healthy subject and w.r.t. to genetic variations in the dopamine pathway, to two small groups of patients with lesions in the basal ganglia and OFC and a group of healthy age-matched controls. Computationally, they use the Experience-weighted attraction model, which nests reinforcement learning and belief-learning, to explain the choice behavior of their participants.

This is an interesting paper that combines cognitive modeling of an established task with specific lesion patient to explore if computational modeling can detect and explain differences in choices in these patient groups. Modeling is done with careful attention to details (but see the comments below) and includes posterior predictive checks. The paper is well-written, although at times it is quite dense and lacks some information that would make a bit more accessible to the reader. For instance, when the EWA model is first described in the Results, it would be helpful to explain the model in a bit more detail than just saying that it nests RL and FP and has been very successful.

However, my biggest concern of this study is sample size. With just 6 patients in each group this study is seriously underpowered to warrant the conclusions that are drawn from these data. I would want to see sample size reaching around 20 patients in each group before I start trusting these results. I do realize that circumscribed strokes in the basal ganglia are rare, but it should not be too difficult to find additional TBI patients with OFC lesions that would fit inclusion criteria. I find it also rather curious that the control subjects were matched w.r.t. to age (and there even no that closely in the BG group). It seems to me that the matching should rather be done w.r.t to estimated intelligence or as a coarse proxy to years of education.

I disagree with the authors that comparison between groups should be only focussed on model comparison indices (pseudo-R2 and BIC) and should not involve actual model parameters. If I see that one model fits better than another, I would want to know why and model parameters can give some insights in the cognitive processes that might be causing this. Also, in contrast to the claim of the authors, I don't see evidence for an inflation of variance in the parameters in the patient groups in Table S2. Rather, I am more concerned with using Maximum Likelihood estimation for fitting the model as this frequently lead to parameter estimates at boundaries of the parameter range, a problem that appears to be present in the BG group for the RL model in the non-strategic condition. Bayesian estimation techniques using MCMC sampling often provide more reliable estimates in such cases.

Overall, the entire paper is strangely focussed on spared strategic learning in the BG group, a group that performed essentially normal during the strategic condition which is the focus of this paper. In contrast, the OFC group show much more impairments in the strategic condition (e.g. Fig. 2B and 2C), and therefore exploring the OFC group in a much more granular fashion would provide more information about the cognitive processes that lead to impaired strategic learning.

Reviewer #3 (Remarks to the Author):

In this paper Zhu et al. (Zhu hereafter) attempt to ascertain the causal contribution of the basal ganglia (BG) to learning in social situations. In non-social games lesion evidence points to a causal contribution of the BG to learning, evidence that agrees with findings from imaging studies. In the social realm imaging points to BG involvement in social learning, but causal evidence has been lacking.

They approach this question by having BG lesion patients, orbital frontal cortex (OFC) lesion patients, and healthy controls (HC) play a social game (and a version without social context), and then use three quantitative approaches to address the question.

First, they use basic statistics and logistic regression across the groups and conditions (social/non-social) to examine the influence of payoff and regret (payoff of optimal choice given opponent's choice - actual payoff). They see similar results both ways: in the social condition both HCs and BGs are more likely to stay with high payouts, while OFCs show no influence of payoff. Conversely, both HCs (and possibly to a greater extent) BGs are less likely to stay with high regret, while OFCs are slightly more likely to stay. In the non-social condition all the groups show a slight positive influence of Payoff, but there is a mixed (across groups), small response to regret. The response to regret is taken as evidence of non-RL (e.g. belief-based) type learning. Hence this ability (measured this way) is still intact (even enhanced?) in the BG patients.

In the second and third approaches Zhu uses computational models. In particular they use EWA, experiential weighted attraction, which parametrically interpolates between pure RL and belief-based play (which is accomplished by updated not only received rewards, but forgone rewards). In the second approach Zhu examines the pseudo - R^2 of the EWA model; in the third they look at the difference between the BIC of the EWA model, and the BIC of the pure RL model. All of these quantities are examined across the groups and conditions. In agreement with what was found using logistic regression, in the pseudo - R^2 case, there is no difference in pseudo - R^2 between HC and BG, but there is between HC and OFC in the strategic condition, but there is a difference between HC and BG and OFC in the non-strategic condition. Similarly, using the differential BIC measure of EWA versus pure RL, in the strategic condition the HC and BG cohorts were significantly bigger than zero, but OFC not; in the non-strategic condition only the OFC group was different than zero and the estimate was negative (RL better).

Overall Impression

This is an impressive analysis of very interesting data on an important topic. Taken all together the evidence presented does a good job supporting the claims made about the contribution of BG to social learning.

Points

1. The title is "Preserved capacity in economic games ..." I would argue there is only one economic game and change the title to "Preserved capacity in an economic game ..."
2. Is the pseudo - R^2 analysis done at the subject level that the error bars reflect the SEM with subjects as the unit of variability? Also, there are numerous pseudo - R^2 's -- please explain this one.
3. For the BIC analysis it is unclear (in light of 2) what was done. Was this analysis done with data pooled over cohort in each condition (which would require a bootstrap type maneuver to get error bars)? I think slightly more explicit explanation of the statistical methods is needed.
4. The focus on BIC is understood, but it would still be interesting to report the EWA parameters (the δ 's especially) estimated from each cohort. More generally it would be interesting to estimate the parameters by subject (perhaps using a Bayesian model to take advantage of partial pooling) and see if they form clusters corresponding to the lesion categories.

5. Line 303: "raises" should be "raise"
6. Line 223 - these results are not reported?
7. Line 190, for example. There are too many "Specifically .."
8. Line 183 "is" should be "are"

Reviewer #1:

Zhu and colleagues studied decision-making in three populations of individuals, patients with basal ganglia (BG) lesions, patients with orbitofrontal cortex (OFC) lesions, and healthy controls. Their central aim was to determine why patients with OFC lesions but not patients with BG lesions often display deficits in social decision-making, when many of the known BG functions should be also important for underlying mechanisms of decision-making. Therefore, this is a highly interest line of inquiry with a broad interest for the field of decision neuroscience.

In order to tackle this question, the authors employed strategic and non-strategic versions of a competitive game known as the Patent Race. While the strategic version of this game involved a mix of both belief-based and reinforcement-based learning, the non-strategic version involved reinforcement-based learning more exclusively. While OFC, but not BG, patients displayed deficits relative to healthy controls in the strategic version of the task, BG patients displayed deficits only in the non-strategic version of the task, suggesting that patients with BG lesions have a dissociable deficit in reinforcement-based learning that can be potentially compensated in the strategic context by intact belief-based learning. This is an interesting and important finding that helps toward clarifying why patients with BG lesions often do not show social decision deficits despite the fact that BG is involved fundamental aspects of learning and decision-making in general. The finding is robust, including nice analyses of behavior from a unique set of patient populations combined with sophisticated modeling. I only have a couple of comments with a goal of further improving the manuscript and clarifying less-explained results. In sum, this is an exciting finding and also a broadly important finding on how BG and OFC contributes to decision-making.

Reply: We thank the reviewer for the positive words!

1. It is clear that the conceptual framework provided for the interpretation of BG lesions is interesting and is in nice correspondence with the results observed. The results concerning OFC lesions, however, warrant some more clarifying explanation from the authors under the same framework. The authors suggest that in order to perform well in the strategic version of their task, people need either belief or reinforcement-based learning to be intact. For the non-strategic version of their task, only reinforcement-based learning is relevant, so only this needs to be intact for performance to be adequate. That logic fits the BG lesion results very well - BG patients have trouble on the task that only requires reinforcement-based learning, but can perform adequately on the strategic version of the task because their belief-based learning is intact. But, according to this framework (i.e., OFC may be involved in both belief and reinforcement-based processes), it is plausible that the OFC patients may also have some trouble in the non-strategic task that only requires the reinforcement-based system. However, at the behavioral level, the OFC lesion group showed intact (and strong) low-high payoff relationship in the non-strategic task (Fig. 2b,c). On the other hand, the section in the Results outlining the Experience-Weighted Attraction (EWA) model found that the OFC lesions led to similarly low pseudo-R2 as the BG group (Fig. 3a) (although Fig. S5 results become much closer to Fig. 2b's data). Some effort was made to explain these findings at the end of the paragraph starting on line 215, but more explanation could help clarify and consolidate these

findings. Basically, I think the authors may want to add more explanation so that consolidating across the behavioral and modelling results become clearer for the readers.

Reply: We thank the reviewer for the careful reading of the manuscript and for bringing up the difference between model-free and model-based findings related to the performance of OFC cohort in the non-strategic condition. One likely reason for this apparent inconsistency is that the stay-switch measure is relatively coarse and does not sufficiently capture all of the potential learning deficits associated with OFC damage. For example, the frequency of switching or staying given payoff/regret does not capture whether participants were switching to more or less adaptive strategies. This suggests that one way to attempt to reconcile the difference between our model-free and model-based results is that although OFC patients were sensitive to payoffs in the non-strategic condition (i.e. switched more often when received low payoffs), their ability of picking the ‘best’ action might have been impaired.

To assess this hypothesis, we calculated the percentage of choosing optimal actions by each cohort under each condition, based on two model-free measures of optimal choices. The first measure involves identifying the best decision based on the history of payoff. Specifically, by comparing the average of received payoffs associated with each action within the past 10 trials, we calculated the proportion of trials in which actions with the highest average payoff were selected. A similar measure was also derived for actions with lowest regret.

Consistent with this hypothesis, we found that OFC patients were indeed less likely to choose the optimal strategy. Specifically, we found that OFC patients performed worse than HC in

both strategic and non-strategic conditions as evaluated by either payoff- (strategic: $44.29 \pm 1.38\%$ vs. $33.16 \pm 2.43\%$, $P < 0.001$; non-strategic: $42.32 \pm 1.38\%$ vs. $33.16 \pm 2.43\%$, $P = 0.001$) or regret-based measure (strategic: $45.67 \pm 1.38\%$ vs. $31.55 \pm 2.40\%$, $P < 0.001$; non-strategic: HC: $39.52 \pm 1.36\%$, OFC: $24.06 \pm 2.21\%$, $P < 0.001$). We have now included these points in the manuscript and the figure in the supplementary materials (Fig. S3).

2. In Fig. 3b, time courses are shown for the pseudo-R2 value for consecutive trials. While pseudo-R2 is fairly consistent over time in the strategic task, this value seems to change noticeably over time in the non-strategic task, especially for the BG patients (rounds 15-35 vs. the rest where it stabilizes). Please provide further explanation either in the results or discussion sections since it suggests some shift in the BG patients' behaviors occurring relatively early in the game.

Reply: We thank the reviewer for pointing out this feature of Fig. 3B, which we agree is quite salient and noteworthy. It is indeed possible that there was a discrete change in the behavior of BG patients in the non-strategic condition. For example, it may be that as in the strategic condition, BG patients were able to compensate (albeit less successfully) in the non-strategic condition by engaging additional cognitive mechanisms, for example by relying on working memory (WM) systems. Indeed, past studies of reward learning suggest that prediction errors produced by RL systems may have a significant contribution from WM systems, especially during early learning as observed in our data (Collins & Frank, 2012; O'Reilly & Frank, 2006).

However, testing the idea of possible time-varying strategies proved challenging. Indeed, even capturing the time series properties was quite difficult. We attempted a variety of different time series measures to ask if we could detect “change-points” in the time series. Ultimately, the results varied considerably depending on the specific assumptions used (e.g., parametric vs. non-parametric trend tests). Given this observation, we have opted for a qualitative description of this feature and possible explanations in the manuscript. At the same time, we welcome any suggestions from the reviewer in terms of both analytical approaches as well as alternative interpretations.

Interestingly, although pseudo- R^2 values were fairly consistent over time in the strategic condition, they were more variable in the non-strategic condition. This is particularly true for the BG patients during rounds 15-35, where pseudo- R^2 dropped sharply following a rise at the start of the experiment. This may reflect the engagement, albeit less successfully than in the strategic condition, of compensatory mechanisms in BG patients in the non-strategic condition, for example through relying on working memory (WM) systems. Indeed, past studies of reward learning suggest that prediction errors produced by RL systems include significant contribution from WM systems, especially during early learning (Collins & Frank, 2012; O'Reilly & Frank, 2006). Future studies are needed to more firmly establish this effect and the underlying neural mechanisms.

- Collins, A. G. E., & Frank, M. J. (2012). How much of reinforcement learning is working memory, not reinforcement learning? A behavioral, computational, and neurogenetic analysis. *European Journal of Neuroscience*, 35(7), 1024–1035.
- O'Reilly, R. C., & Frank, M. J. (2006). Making Working Memory Work: A Computational Model of Learning in the Prefrontal Cortex and Basal Ganglia. *Neural Computation*, 18(2), 283–328.

Reviewer #2:

In this paper, Zhu et al. administered an economic game, called the Patent Race, which they have previously investigated in healthy subject and w.r.t. to genetic variations in the dopamine pathway, to two small groups of patients with lesions in the basal ganglia and OFC and a group of healthy age-matched controls. Computationally, they use the Experience-weighted attraction model, which nests reinforcement learning and belief-learning, to explain the choice behavior of their participants.

1. This is an interesting paper that combines cognitive modeling of an established task with specific lesion patient to explore if computational modeling can detect and explain differences in choices in these patient groups. Modeling is done with careful attention to details (but see the comments below) and includes posterior predictive checks. The paper is well-written, although at times it is quite dense and lacks some information that would make a bit more accessible to the reader. For instance, when the EWA model is first described in the Results, it would be helpful to explain the model in a bit more detail than just saying that it nests RL and FP and has been very successful.

Reply: We thank the reviewer for the comments and suggestions. We include an expanded description of the EWA model in the supplementary materials and now refer to it in the main text:

Fig. S1. Illustration of behavior and model predictions using data from representative healthy participants in the Weak role (i.e., can invest from 0 to 4). Actual time series of choice was plotted by using a 15-round bin average. Choice probabilities were generated using the best-fitting parameters based on the RL and hybrid EWA learning, respectively. (A)

Strategic condition: The engagement in the belief learning under the strategic

condition, which is not captured by RL, is most saliently reflected in a decreased probability of investing 0 and increased probability of 4 in rounds 40-60. This corresponding to periods when Strong players invested 1-2 units with increased likelihood. EWA model fit demonstrates a significant improvement over RL, which fails to account for the strategy shift during this period (BIC difference between EWA and RL = 9.69; EWA pseudo- $R^2 = 0.33$). (B) Non-strategic condition: Both RL and EWA captured the overall pattern of the observed choice dynamics, yielding almost identical sequences of predictions. Formal model comparison suggested that, whereas EWA provided a good fit for the data (EWA pseudo- $R^2 = 0.70$), it fit worse than RL after penalizing for the additional parameters (BIC difference

between EWA and RL = -4.93), consistent with previous findings suggesting learning was primarily driven by reinforcement learning when the information of incentive structure of the opponent was missing.

2. However, my biggest concern of this study is sample size. With just 6 patients in each group this study is seriously underpowered to warrant the conclusions that are drawn from these data. I would want to see sample size reaching around 20 patients in each group before I start trusting these results. I do realize that circumscribed strokes in the basal ganglia are rare, but it should not be too difficult to find additional TBI patients with OFC lesions that would fit inclusion criteria. I find it also rather curious that the control subjects were matched w.r.t. to age (and there even no that closely in the BG group). It seems to me that the matching should rather be done w.r.t to estimated intelligence or as a coarse proxy to years of education.

Reply: We thank the reviewer for raising these two distinct issues: sample size of our lesion cohorts, and matching of lesion and healthy comparison (HC) cohorts. We address each point in turn.

On the former, we agree with the reviewer that sample size is indeed an important consideration. The sample size choices for our study were informed by the following: (i) previous lesion studies in cognitive neuroscience, (ii) availability of patient access, and (iii) power calculations.

First, our study, with 6 patients in each patient cohort, is in keeping with previous human lesion studies in the cognitive neurosciences generally (Feinberg et al., 2000; Szczepanski and Knight, 2014). Within neuroeconomics specifically, Koenigs et al. (2007) for example examined 6 patients with vmPFC lesions; Zhu et al. (2014) tested 6 patients with DLPFC damages and 7 with OFC damages; Gu et al. (2015) studied vmPFC and insula cohorts with the sample size of 6 and 7, respectively. That lesion effects on cognition can be detected with small sample sizes is not surprising since reliable effects are obtained in monkey experiments with only 1 or 2 subjects if the neuroanatomy of the lesion (and/or reversible inactivation) is well controlled. Indeed, comparable group sizes have been shown to produce robust behavioral effects across all areas in the cognitive neurosciences, ranging from memory, attention, language, and motor systems (for extensive reviews of this literature, see Feinberg et al., 2000 and Szczepanski and Knight, 2014).

Second, because the primary focus of our study is on basal ganglia functioning, our statistical power is largely determined by the sample size of the BG patient cohort, which, as noted by the reviewer, is inherently limited by the rarity of patients with circumscribed stroke in the basal ganglia. Increasing the sample size of the OFC lesion cohort will not address this issue. Indeed, the rarity of these patients is an important reason for the paucity of information on the causal contribution of BG relative to regions such as OFC, and for our focus on this structure in a balanced experimental design.

Third, our choice of sample sizes was also guided by the power analysis that evaluated the sensitivity of our design using a model-based simulation approach (Muthén et al., 2002; Cohen 1988; Zhu et al., 2014). Specifically, we evaluated the power of detecting a difference in pseudo- R^2 of EWA model fit between HC and lesion patients who were assumed to behave randomly. Because our pilot HC data indicated that values of pseudo- R^2 for the best-fitting EWA were 0.43 and 0.45 in the strategic and non-strategic condition respectively, with standard deviation of .2 for both, we conducted our power analysis using the same mean estimates for HC and 0 (pseudo- R^2 for random choice model) for patients, assuming the same estimates of .2 as standard deviation for both cohorts. Our computation, under alpha level .05 and power $(1-\beta) = .8$, indicated a sample size of 4 patients is adequate to detect the difference in model fit where they exist.

We also thank the reviewer for bringing up issues related to the matching between patients and controls on demographic and cognitive factors, including age, intelligence, and years of education. Regarding age, this was included in Fig. S5 of the original submission. Specifically, we show that partitioning our HC cohort, into age-matched BG and OFC controls in both mean and standard deviation, yielded similar results as those using the combined HC cohort.

In addition, following the reviewer's suggestion, we conducted additional statistical tests for the effect of intelligence and education. We found that behavioral performance was not significantly correlated with either intelligence or years of education in any task (Strategic: IQ: Spearman $\rho=0.02$, $P=0.918$; years of education: $\rho=-0.07$, $P=0.711$; Non-strategic: IQ: $\rho=0.08$, $P=0.664$; year of education: $\rho=0.09$, $P=0.607$) or in any cohort (IQ: HC: $\rho=-0.17$, $P=0.296$, BG: $\rho=0.16$, $P=0.629$, OFC: $\rho=0.44$, $P=0.154$; year of education: HC: $\rho=-0.14$, $P=0.384$, BG: $\rho=0.27$, $P=0.405$, OFC: $\rho=0.03$, $P=0.918$), and the reported results were robust to the inclusion of demographic variables, including age, gender, IQ, and years of education, as nuisance covariates (EWA pseudo- R^2 : strategic: HC vs. BG: $\beta=-0.01\pm 0.03$, $P=0.725$, HC vs. OFC: $\beta=0.33\pm 0.03$, $P<0.001$; non-strategic: HC vs. BG: $\beta=0.13\pm 0.03$, $P<0.001$, HC vs. OFC: $\beta=0.17\pm 0.03$, $P<0.001$). We have now included these results in the *Supplementary Tables (Table S2 and Table S4)*.

- Feinberg, Todd E., and Martha J. Farah, eds (2000). Patient-based approaches to cognitive neuroscience. MIT Press.

- Szczepanski, S.M., Knight, R.T., (2014). Insights into Human Behavior from Lesions to the Prefrontal Cortex. *Neuron*, 83, 1002–1018.
 - Koenigs, M., Young, L., Adolphs, R., Tranel, D., Cushman, F., Hauser, M., & Damasio, A. (2007). Damage to the prefrontal cortex increases utilitarian moral judgements. *Nature*, 446(7138), 908–911.
 - Zhu, L., Jenkins, A.C., Set, E., Scabini, D., Knight, R.T., Chiu, P.H., King-Casas, B., Hsu, M., (2014). Damage to dorsolateral prefrontal cortex affects tradeoffs between honesty and self-interest. *Nat Neurosci*.
 - Gu, X., Wang, X., Hula, A., Wang, S., Xu, S., Lohrenz, T. M., et al. (2015). Necessary, Yet Dissociable Contributions of the Insular and Ventromedial Prefrontal Cortices to Norm Adaptation: Computational and Lesion Evidence in Humans. *Journal of Neuroscience*, 35(2), 467–473.
 - Muthén L. K. et al. (2002), How to Use a Monte Carlo Study to Decide on Sample Size and Determine Power, *Structural Equation Modeling: A Multidisciplinary Journal*, 9, 599–620.
 - Cohen, J. (1988), *Statistical Power Analysis for the Behavioral Sciences*, 2nd ed. Academic Press, New York.
3. I disagree with the authors that comparison between groups should be only focussed on model comparison indices (pseudo-R2 and BIC) and should not involve actual model parameters. If I see that one model fits better than another, I would want to know why and model parameters can give some insights in the cognitive processes that might be causing this. Also, in contrast to the claim of the authors, I don't see evidence for an inflation of variance in the parameters in the patient groups in Table S2. Rather, I am more concerned with using Maximum Likelihood estimation for fitting the model as this frequently lead to parameter estimates at boundaries of the parameter range, a problem that appears to be present in the BG group for the RL model in the non-strategic condition. Bayesian estimation techniques using MCMC sampling often provide more reliable estimates in such cases.

Reply: We thank the reviewer for raising this issue and concur that we provided an imprecise explanation in our initial submission. We used goodness-of-fit as our key criteria due to concerns regarding model misspecification with lesion cohort behavior, as model-based approaches such as RL or EWA inherently make strong assumptions regarding how past experiences are integrated over the course of learning.

Specifically, comparison of parameters provides meaningful insights into the cognitive components underlying behavior if both cohorts behave in accordance to model assumptions, but differ in the weighting of specific components. For example, using comparison between HC strategic and non-strategic conditions, the belief learning parameter (δ) provided a good indication that HC relied less on belief learning in the non-strategic condition ($.7 \pm .38$ in the strategic condition, and $.45 \pm .47$ in the non-strategic condition), a result that was consistent with model-free findings. On the other hand, if the model fit is significantly poorer for a particular cohort, it suggests that one or more model assumptions were violated. This misspecification

issue is equally true for both frequentist and Bayesian approaches. More importantly, this model misspecification can result in either upward or downward biases in parameter estimates. This makes it difficult, even in the presence of significant differences in parameter estimates, to draw firm conclusions regarding differences in cognitive mechanisms between cohorts or conditions.

This concern regarding model misspecification is the reason we devoted substantial attention to model-free analyses, which makes fewer assumptions regarding the specific way in which past experiences are integrated to guide decisions. Specifically, model-free results provide some suggestion that both BG and HC cohorts were sensitive to past payoffs and regret in the strategic condition, but that BG patients were not sensitive to payoffs and regret in the non-strategic condition. In contrast, OFC patients differed in that they did not respond to either past payoff or regret in strategic condition. We have now clarified this point in the manuscript:

A more general concern with our model-based approach is the possibility of model misspecification due to participants engaging in decision rules beyond the EWA model space⁵⁷. This is particularly the case with lesion cohort behavior, as model-based approaches such as RL or EWA inherently make strong assumptions regarding how past experiences are integrated over the course of learning. Our study addressed this in two ways. First, we focused on cross-cohort comparisons using goodness-of-fit measures, rather than specific parameters calibrated from the behavior (e.g., the weight on belief-based learning). Although comparisons based on individual parameters provide meaningful insights into the cognitive components underlying behavior if cohorts behave in accordance to model assumptions, they can result in serious inferentially errors when the underlying assumptions are violated⁵⁸. For example, using comparison between HC strategic and non-strategic conditions, the belief learning parameter provided a good indication that HC relied less on belief learning in the non-strategic condition. On the other hand, a poor model fit raises the possibility that one or more model assumptions are violated. This misspecification issue is equally true for both frequentist and Bayesian approaches. More importantly, such model misspecification can result in either upward or downward biases in parameter estimates. This makes it difficult, even in the presence of significant differences in parameter estimates, to draw firm conclusions regarding differences in cognitive mechanisms between cohorts or conditions.

Second, consistent with other neuroeconomic studies using the lesion method (e.g., Noonan et al., 2017; Peters & D'Esposito, 2016; Gu et al. 2015; Zhu et al., 2014; Kovach et al., 2012), we used model-free analyses to characterize behavioral deficits and support conclusions derived from model-based methods. Specifically, this involved examining the extent to which participants' choice behavior was sensitive to various different notions of learning signals without restricting the specific functional form that weights on these signals may take (Fig. S10). A more thorough investigation using data-driven approaches will be needed to further assess and compare choice predictability of lesion patients vs. control subjects, removing assumptions regarding which and how external stimuli drive the learning process.

- Noonan, M. P., Chau, B. K. H., Rushworth, M. F. S., & Fellows, L. K. (2017). Contrasting Effects of Medial and Lateral Orbitofrontal Cortex Lesions on Credit Assignment and Decision-Making in Humans. *Journal of Neuroscience*, 37(29), 7023–7035.
 - Peters, J., & D’Esposito, M. (2016). Effects of Medial Orbitofrontal Cortex Lesions on Self-Control in Intertemporal Choice. *Current Biology*, 26(19), 2625–2628.
 - Gu, X., Wang, X., Hula, A., Wang, S., Xu, S., Lohrenz, T. M., et al. (2015). Necessary, Yet Dissociable Contributions of the Insular and Ventromedial Prefrontal Cortices to Norm Adaptation: Computational and Lesion Evidence in Humans. *Journal of Neuroscience*, 35(2), 467–473.
 - Zhu, L., Jenkins, A.C., Set, E., Scabini, D., Knight, R.T., Chiu, P.H., King-Casas, B., Hsu, M., (2014). Damage to dorsolateral prefrontal cortex affects tradeoffs between honesty and self-interest. *Nat Neurosci*.
 - Kovach, C. K., Daw, N. D., Rudrauf, D., Tranel, D., O’Doherty, J. P., & Adolphs, R. (2012). Anterior Prefrontal Cortex Contributes to Action Selection through Tracking of Recent Reward Trends. *Journal of Neuroscience*, 32(25), 8434–8442.
4. Overall, the entire paper is strangely focused on spared strategic learning in the BG group, a group that performed essentially normal during the strategic condition which is the focus of this paper. In contrast, the OFC group show much more impairments in the strategic condition (e.g. Fig. 2B and 2C), and therefore exploring the OFC group in a much more granular fashion would provide more information about the cognitive processes that lead to impaired strategic learning.

Reply: We appreciate the reviewer for highlighting the importance of OFC contributions. We do, however, disagree that our focus on the BG is “strange”. First, as we highlighted in the introduction, much more is known about the causal role of OFC to social and strategic decision-making in comparison to BG. Indeed, based on the existing literature of the extensive decision-making deficits observed following OFC lesion, it would be surprising if OFC damage was not associated with impairments in our study.

Second, also highlighted in the introduction, there is the matter of the apparent discrepancy between the neuroimaging and clinical evidence of BG involvement in social and strategic decisions. Accordingly, our study aimed to fill the critical gap between the accumulating correlational findings suggesting the involvement of BG in social decision-making and the absence of causal evidence, that either in support or argue against, this involvement. Thus, the goal of our study is not so much as to document and understand functional impairments but to test different hypotheses regarding the impairment and sparing of selective cognitive functions. A more granular understanding of OFC impairments, while certainly worthwhile and important in its own right, is beyond the scope of the present paper.

Reviewer #3

Overall Impression

This is an impressive analysis of very interesting data on an important topic. Taken all together the evidence presented does a good job supporting the claims made about the contribution of BG to social learning.

Reply: We thank the reviewer for positive words regarding our manuscript.

Points

1. The title is "Preserved capacity in economic games ..." I would argue there is only one economic game and change the title to "Preserved capacity in an economic game ..."

Reply: We thank the reviewer for this suggestion, and have now changed the title to "Preserved capacity to learn in an economic game following basal ganglia damage".

2. Is the pseudo - R^2 analysis done at the subject level that the error bars reflect the SEM with subjects as the unit of variability? Also, there are numerous pseudo - R^2 's -- please explain this one.

Reply: We thank the reviewer for urging us to clarify the calculation of pseudo- R^2 and its SEM. Following Camerer and Ho (1999), the pseudo- R^2 was defined as the difference between the log likelihood of the EWA model and a random choice model, scaled by log likelihood of the random-model. The means and error bars were constructed using a bootstrap procedure pooling over cohorts for each condition. Specifically, on each iteration, we constructed a pseudo-sample of a specific cohort and condition by resampling, with replacement, the pseudo- R^2 of all trials across all subjects within that cohort. We now include this information in the manuscript.

- **Camerer, C.F., Ho, T., (1999). Experienced-weighted attraction learning in normal form games. *Econometrica* 67, 827-874.**

3. For the BIC analysis it is unclear (in light of 2) what was done. Was this analysis done with data pooled over cohort in each condition (which would require a bootstrap type maneuver to get error bars)? I think slightly more explicit explanation of the statistical methods is needed.

Reply: We now include additional explanation for the BIC comparison. BIC was calculated by pooling over participants in each cohort, and dividing by the sample size of the cohort to account for differences in number of participants in each cohort. BIC error bars were calculated using a bootstrap procedure pooling over cohorts across conditions. Specifically, on each iteration, we constructed a pseudo-sample of a specific cohort and condition by resampling, with replacement, the log-likelihood

values of all trials across all subjects within that cohort, and penalizing the additional number of parameters to derive the BIC. This has now been included in the manuscript.

4. The focus on BIC is understood, but it would still be interesting to report the EWA parameters (the δ 's especially) estimated from each cohort. More generally it would be interesting to estimate the parameters by subject (perhaps using a Bayesian model to take advantage of partial pooling) and see if they form clusters corresponding to the lesion categories.

Reply: We thank the reviewer for the comment. Following reviewer suggestion, we now highlight the parameter estimates, originally reported in Table S3, in the revision. The parameter δ , for example, was estimated to be $.70 \pm .38$ for HC and $.70 \pm .46$ for BG in the strategic condition, and $.45 \pm .47$ for HC in the non-strategic condition, consistent with findings suggesting a reduced reliance on belief learning (thus smaller δ) when the information about the strategic context is absent.

Note that although we reported parameter estimates of each cohort under each condition, caution is needed when interpreting estimates for cohorts with poor model fits (as in the case of BG non-strategic, OFC strategic and non-strategic). This is because, when the model fit is significantly poorer for a particular cohort, it is possible that one or more model assumptions are violated, which can result in either upward or downward biases in parameter estimates. This model misspecification issue exists regardless of whether frequentist or Bayesian approach was used for estimation (see also our response to Reviewer 2, point 3).

Second, we thank the reviewer for the excellent suggestion of classifying subjects' lesion labels (BG vs. OFC vs. HC) based on inputs derived from EWA estimation. Rather than using a completely unsupervised clustering methods, however, we fit and compared 3 multinomial logistic regression models that predict individual labels. We chose this method in order to compare between model inputs, which is straightforward for logistic regression models but much less so for clustering methods such as k-means.

Specifically, in the first model, which serves as a benchmark, we included only an intercept term to account for different sample sizes across cohorts. The second model, which serves to test the predictive power of EWA parameters, contained in addition the estimated parameters of each individual based on the best fitting EWA. The third model, which serves to test whether information related to model fits can predict brain lesion category above and beyond parameter estimates, contained in addition the value of pseudo- R^2 for each individual.

Using likelihood ratio tests, we compared the explanatory power of the 3 models in each condition. In both strategic and non-strategic conditions, Model 2 (intercept + EWA parameters) was not able to predict brain lesion categories above and beyond the benchmark Model 1 (strategic:

$\chi^2(10) = 15.79, P = 0.106$; non-strategic: $\chi^2(10) = 10.94, P = 0.362$). However, after including individual pseudo-R² values, the clustering showed consistently better recovery of individual lesion labels than when using benchmark (strategic: $\chi^2(12) = 25.71, P = 0.012$; non-strategic: $\chi^2(12) = 29.12, P = 0.004$) or parameter estimates alone (strategic: $\chi^2(2) = 9.92, P = 0.007$; non-strategic: $\chi^2(2) = 18.18, P < 0.001$).

In addition, using a leave-one-out procedure based on the winning model to evaluate the classification performance for each cohort under each condition, the overall out-of-sample classification accuracy was .61.

Interestingly, and consistent with findings that BG patients performed similarly to HC in the strategic condition, the classifier did particularly poorly in distinguishing between BG and HC in the strategic environment, mislabeling 5 out of 6 BG patients as HC. On the other hand, in the non-strategic condition, the model made fewer mistakes in distinguishing between HC and the lesion patients, consistent with findings that HC outperformed both patient cohorts in the non-strategic environment.

		Strategic			Non-strategic		
		HC	BG	OFC	HC	BG	OFC
LOO classification	OFC	2	1	4	0	2	3
	BG	4	0	0	4	2	1
	HC	14	5	2	16	2	2
		HC	BG	OFC	HC	BG	OFC
		Group			Group		

We have now included these results in the Supplement (Fig. S9)

- Line 303: "raises" should be "raise"

Reply: We have corrected the typo.

- Line 223 - these results are not reported?

Reply: We have now included results of self-tuning EWA in the Supplement (Fig. S8), replicated below.

To evaluate the robustness of our results, we estimated self-tuning EWA model, which contains only two free parameters (i.e., inverse temperature and learning rate) reducing all other learning parameters into either fixed values or functions of

experience (Ho et al., 2007). While the estimation results were largely consistent with that based on the standard EWA (as shown in Fig. 3A), the difference between OFC and HC cohorts in the non-strategic condition ceased to be insignificant, suggesting that OFC effects were sensitive to alternative model specifications.

7. Line 190, for example. There are too many "Specifically .."

Reply: We have rephrased the sentence as suggested by the reviewer.

8. Line 183 "is" should be "are"

Reply: We have corrected the mistake.

****REVIEWERS' COMMENTS:**

Reviewer #1 (Remarks to the Author):

The authors addressed all of my concerns. I have no further issues and appreciate that the authors performed additional analyses to convey their points well.

(Additional request from the Editor): "We also request that you could comment on whether the authors have adequately addressed the sample size concern raised by Reviewer 2."

I believe that the authors' responses to the sample size issue for R2 was satisfactory. They point out that the current study's sample size does not deviate much from the typical sample size used for this type of lesion studies in humans. Also, they point out the practical limitation of studying a large number of BG lesion patients. Even in pre-planned monkey lesion studies (say, amygdala), n of 4 can be more than sufficient for studying behavioral effects due to a specific brain lesion. Finally, the authors provide that the sample size was guided by their power analysis for obtaining effects for their main finding, which helps. Overall, I think there will always be some sample size limitation when studying human patients with relatively selective lesions to specific brain areas. Given that their main findings show robust effects, I am convinced that the current finding adds valuable knowledge based on the special patient populations.

Reviewer #2 (Remarks to the Author):

The authors have responded to in detail to my satisfaction to most of my comment. The only issue that remains for me is that I am not convinced by the arguments that the study was appropriately powered. While it is true that previous lesion studies have also used sample sizes of 6-7 patients in one group, this doesn't make the present study more valid, because the actual power that a study has is determined by the effects size, which was much bigger than in the current case.

While I do appreciate the authors attempt for a power estimation based on pilot data from healthy controls (why are these not included in the Supplement?) I think that they are overestimating the expected effect size in their favor. It is quite unrealistic that patients would behave completely random and hence the expected pseudo-R2 would be zeros. Under these unrealistic assumptions the author are correct that a sample size of 4 would be sufficient for a statistical power of 80%. However, if I plug in the real pseudo-R2 for the BG group (0.25 as stated in the legend of Figure S5), the actual statistical power of this comparison drops to 44%. To achieve a power of 80%, 15 BG patients and 30 HC participants would have to have been tested. While of course the true effect size cannot be know ahead of the study, the assumptions that went into the power calculations were unrealistically optimistic. i think the authors should reconsider their expectations about the model fit in the BG group or even better try to find a few more BG patients that would bolster their conclusions.

Reviewer #3 (Remarks to the Author):

The authors have answered a wide range of queries from three reviewers including me. Their answers have completely clarified my questions.

Their responses were detailed and focused on the questions being asked.

This is an excellent contribution and i would like to commend the authors on their response to reviews.

Reviewer #2:

The authors have responded to in detail to my satisfaction to most of my comment. The only issue that remains for me is that I am not convinced by the arguments that the study was appropriately powered. While it is true that previous lesion studies have also used sample sizes of 6-7 patients in one group, this doesn't make the present study more valid, because the actual power that a study has is determined by the effects size, which was much bigger than in the current case.

While I do appreciate the authors attempt for a power estimation based on pilot data from healthy controls (why are these not included in the Supplement?) I think that they are overestimating the expected effect size in their favor. It is quite unrealistic that patients would behave completely random and hence the expected pseudo-R² would be zeros. Under these unrealistic assumptions the author are correct that a sample size of 4 would be sufficient for a statistical power of 80%. However, if I plug in the real pseudo-R² for the BG group (0.25 as stated in the legend of Figure S5), the actual statistical power of this comparison drops to 44%. To achieve a power of 80%, 15 BG patients and 30 HC participants would have to have been tested. While of course the true effect size cannot be know ahead of the study, the assumptions that went into the power calculations were unrealistically optimistic. i think the authors should reconsider their expectations about the model fit in the BG group or even better try to find a few more BG patients that would bolster their conclusions.

Reply: We appreciate the reviewer for bringing up alternative assumptions for our power calculations. As in all power analyses, it is possible to justify some range of assumptions regarding the calculations. Because effect size estimates are inherently subject to noise, adjudicating between different assumptions in calculations of statistical power can be challenging. This is a well-known issue and, similar to the debate on statistical significance, has resulted in extreme views both in terms of abandoning the statistical concept itself, or insist on some binary classification based on some pre-set value. Instead, we believe that power calculation for a single study is valuable but should not be the sole criteria in determining the validity of a study, hence our reference of the larger lesion literature which we believe provides more robust guidance on sample size and other experimental considerations.

To address the concerns raised by the reviewer, therefore, we have included: (i) the power analysis calculations in the main text, (ii) additional discussion on the sample size limitation and suggestions for possible directions for future studies to overcome this issue, in particular given the inherent rarity of focal BG lesions, and (iii) to help readers make a better judgment about the effect size of our study, confidence intervals for test statistics reported in the paper.